# Electronic and Nuclear Subsystem Response in Hybrid Halide Perovskites Under γ-Irradiation

**DOI:** 10.3390/nano15191474

**Published:** 2025-09-25

**Authors:** Ivan E. Novoselov, Ivan S. Zhidkov

**Affiliations:** 1Institute of Physics and Technology, Ural Federal University, Mira 19 Street, Yekaterinburg 620002, Russia; 2M.N. Mikheev Institute of Metal Physics, Ural Branch of Russian Academy of Sciences, S. Kovalevskoi 18 Street, Yekaterinburg 620108, Russia; 3Federal Research Center for Problems of Chemical Physics and Medicinal Chemistry RAS, Semenov Ave. 1, Chernogolovka 142432, Russia

**Keywords:** lead halide perovskites, radiation effects, Monte-Carlo simulations, GEANT4

## Abstract

Lead halide perovskites, including single-cation (MAPbI_3_, FAPbI_3_, CsPbI_3_) and mixed-cation (Cs_0.12_FA_0.88_PbI_3_, Cs_0.1_MA_0.15_FA_0.75_PbI_3_) compositions, are promising for both space photovoltaics and γ-ray detection due to their tunable optoelectronic properties. However, their response to high-energy radiation remains critical for reliable operation. We performed Monte-Carlo simulations using GEANT4 to investigate photon interactions (0.1–90 MeV) with perovskites of varying composition and thickness (1 cm to 1 μm). Results indicate that heavy atoms (Pb, I) dominate photoelectric absorption and scattering, broadly similar absorbed energies and event rates across compositions. Cs-containing perovskites exhibit slightly higher absorption and ionization, whereas FA- and MA-rich compositions show reduced photoelectric and Rayleigh scattering. Layer thickness strongly influences the radiation response: ultrathin films display fewer interactions with higher per-event energy, while millimeter-scale layers achieve efficient absorption and enable pair-production events at MeV energies. The sequence of dominant processes follows the expected energy dependence: photoelectric effect at low energies, Compton and Rayleigh scattering at intermediate energies, and pair production at high energies. These findings demonstrate that perovskite γ-interaction is primarily governed by heavy-atom content, with A-site cations fine-tuning the process balance, and that device performance for detection or photovoltaics depends critically on layer thickness.

## 1. Introduction

Lead halide perovskites, based on cations such as methylammonium (MA^+^), formamidinium (FA^+^), cesium (Cs^+^), and their mixed compositions, have attracted considerable attention in recent years for their exceptional optoelectronic properties and versatility in applications ranging from photovoltaics to radiation detection [1,2,3]. In particular, their tunable band gaps [4,5], high absorption coefficients [4,6], and long carrier diffusion lengths [7] make them promising candidates for space applications [8]. However, the long-term stability of perovskites under high-energy radiation [9,10] remains a key challenge for both fundamental research and practical deployment.

Despite their promising attributes, perovskites are plagued by instabilities that hinder their widespread adoption. These materials exhibit stress in various environmental conditions, including photodegradation [11,12,13] under prolonged illumination, thermal decomposition [12,13] at elevated temperatures, oxidative reactions [13] in the presence of oxygen or moisture, and degradation induced by ionizing radiation [14]. Among these, radiation stability is particularly critical for applications in harsh environments, such as space-based photovoltaics or radiation detectors, where exposure to cosmic rays and solar particles can deliver intense doses of high-energy particles. γ-rays, one of the prevalent forms of ionizing radiation in such settings, poses a unique threat due to its deep range in matter and ability to induce profound electronic and structural changes at the atomic level.

Exposure to γ-radiation initiates a cascade of physical and chemical processes in perovskite materials. The interaction of high-energy photons with the crystal lattice can lead to the ejection of core and valence electrons, generating a dense population of secondary electrons, excitons, and localized charges [15,16]. In lead-containing halide perovskites, ionization of the Pb core levels (e.g., 4f, 5d) becomes significant when the incident photon energy exceeds the corresponding binding energies [17,18]. These processes can trigger a range of redox-like chemical reactions within the perovskite lattice, including halide migration, oxidation or reduction of lead ions (Pb^0^, Pb^1+^, Pb^2+^), and partial decomposition of organic cations [19].

Interestingly, many of the degradation pathways observed under photon irradiation resemble those induced by intense optical illumination (photodegradation), suggesting a degree of overlap in the underlying mechanisms. In both cases, the generation of high-energy carriers, the formation of defect states, and the involvement of ionic species play crucial roles [20,21,22]. However, the energy deposition channels in γ-irradiation are far more diverse: depending on photon energy, Compton and Rayleigh scattering, the photoelectric effect, and, at even higher energies, pair production can dominate the interaction.

The balance between these processes depends strongly on the γ-ray energy range. At lower energies (tens to hundreds of keV), the photoelectric effect—particularly on high-Z elements such as Pb—is the dominant interaction channel, leading to efficient core-level ionization and localized lattice damage. At intermediate energies, Compton and Rayleigh scattering prevail, producing more delocalized energy deposition and extended carrier tracks. At even higher energies (MeV range), pair production becomes significant, creating energetic electron–positron pairs that further enhance ionization density [16].

Understanding how these mechanisms vary across energy ranges is crucial for tailoring the composition and microstructure of perovskites for radiation hardness. For example, certain compositions or mixed-cation systems may better withstand localized ionization damage in the lower-energy γ regime, while others may exhibit improved resistance against the more distributed defect generation characteristic of high-energy exposure. By mapping the dominant interaction channels and their associated damage pathways, it becomes possible to identify design strategies for more stable perovskite-based solar cells.

Beyond space photovoltaics, lead halide perovskites are also being explored as active media for X-ray and γ-ray detection. Their high absorption coefficients, solution processability, and potential for low-cost, large-area fabrication make them promising candidates for medical imaging, security screening, and dosimetry applications [23,24,25,26,27]. In X-ray detectors, where micrometer-scale films are typically sufficient, the partial transmission of photons through the absorber can be advantageous, enabling high-resolution imaging with controlled attenuation [23,24]. Conversely, for γ-ray detection and spectroscopy, thicker crystals on the millimeter scale are required to ensure efficient absorption and minimize transmission losses [26,27]. These considerations highlight that optimizing perovskite composition and device geometry for radiation detection must take into account not only fundamental interaction mechanisms, but also the balance between absorption depth and application-specific requirements.

Thus, a comprehensive understanding of radiation effects in perovskites requires consideration not only of the primary interaction mechanisms, but also of the secondary phenomena associated with energy transfer—localized heating. In the present study, we employ Monte-Carlo simulations to investigate the processes occurring in perovskite materials under γ-ray exposure, with particular emphasis on the effects it may induce.

## 2. Calculation Details

High-energy particle exposure leads to crystal structure damage and degradation of material properties. In some aspects, γ-irradiation can resemble photodegradation—for example, in the formation of similar defect types—though the underlying mechanisms differ. To investigate the nature of the electromagnetic processes, their impact on defect formation, and the radiation stability of devices in space, both experimental studies and modeling are required.

In this study, several perovskite materials were considered: model single-cation systems (MAPbI_3_, FAPbI_3_, CsPbI_3_) and mixed multi-cation ones (Cs_0.12_FA_0.88_PbI_3_, Cs_0.1_MA_0.15_FA_0.75_PbI_3_). These compounds have well-characterized properties and are widely used in research due to their high energy-conversion efficiency and relatively simple synthesis. Their stability under external factors, including radiation, varies, making them suitable for analyzing degradation processes. Mixed perovskites—complex hybrid structures combining different cations—offer enhanced performance through synergistic effects. Cs_0.12_FA_0.88_PbI_3_ shows high thermal stability due to the inclusion of cesium, making it promising for high-temperature operation, such as in space [28]. Meanwhile, Cs_0.1_MA_0.15_FA_0.75_PbI_3_ combines ease of synthesis, stability, and efficiency, also due to mixed-cation synergy [29]. This selection enables a comparative analysis of radiation effects on both single- and multi-cation perovskite systems.

To model the influence of radiation exposure, a comprehensive simulation approach was employed using the GEANT4 toolkit [30,31]. The study considered γ-rays over a wide energy range (0.1–90 MeV), including reference γ-lines from common calibration sources—^137^Cs and ^60^Co (0.662 MeV, 1.17 MeV, 1.33 MeV) for interaction with different perovskites. GEANT4 enables detailed analysis of particle-matter interactions and damage mechanisms.

Simulations were carried out for thick (1 cm) perovskite targets. While the active layer in perovskite solar cells is typically 200–500 nm, thicker targets or high-flux irradiation are needed to capture statistically significant radiation–matter interactions. GEANT4 can simulate fluxes up to 9.9 × 10^9^ particles/cm^2^, but such high-flux runs are computationally expensive. Using a 1 cm target instead of a 0.5 μm layer increases the number of possible interaction events by a factor of 2 × 10^6^. Therefore, a 1 cm thick target at a flux of 10^5^ particles/cm^2^ yields the same number of events as a 500 nm layer at 10^11^ particles/cm^2^.

After establishing this baseline, the influence of reduced absorber thickness was systematically explored. Simulations were performed for layers of 1 mm, 100 μm, 10 μm, and 1 μm, which represent characteristic dimensions reported in perovskite-based X-ray detectors and medical imaging devices [23,24,25,26,27,32,33,34,35,36]. This selection reflects practically relevant thicknesses spanning from bulk-like absorbers to thin films used in transmission-mode applications.

A detailed description of the simulation setup—including material selection, atomic composition, physics model configuration, and particles’ flow in simulation volume (Appendix A)—is provided in the Appendix A.

## 3. Results and Discussion

### 3.1. Composition Influence on Radiation Processes

The outputs of these GEANT4 simulations include the absorbed energy, non-ionizing energy loss, event rates, and contributions of individual interaction processes for each perovskite composition across a range of photon energies and target thicknesses. These results form the basis for the following analysis, where the similarities and differences between single- and multi-cation perovskites are examined, along with the underlying physical mechanisms governing their γ-radiation response.

From Figure 1a and Table 1, showing the deposit energy as a function of perovskite thickness, there is no significant difference between the materials (additional plots of deposit energy as a function of thickness for other perovskites is provided in Appendix A). This can be explained by the high penetration ability of γ-radiation and its relatively low probability of interaction per unit path length in these compounds.

The dominant interactions occur with heavy atoms in the lattice—primarily Pb and I—which are present in comparable amounts across all studied compositions. Three of the perovskites also contain Cs, which could slightly increase the absorption due to its high atomic number. This is consistent with Figure 1b, where Cs-containing perovskites show marginally higher deposited energy, whereas FA-rich compositions (containing more light atoms) show slightly lower values.

For non-ionizing energy loss (NIEL) shown in Figure 1c, the differences between materials are negligible. This is expected since γ-rays interact predominantly through processes that do not cause significant atomic displacements, passing through the material with minimal momentum transfer to nuclei.

The shape of the absorbed-energy curves is not trivial. For reference materials such as Al, Pb, and PET (See Appendix A), the profiles differ from those of the studied perovskites, which exhibit a pronounced minimum.

This may be related to the interplay of competing processes—the transition from photoelectric dominance to Compton and Rayleigh scattering—and the specific mixture of high-Z and low-Z elements in the perovskites, producing a distinct crossover energy where absorption is minimized.

The number of events per incident particle (Figure 2a–e) and the average energy deposited per event are also very similar among all perovskites. This uniformity is likely due to the similar effective atomic numbers (Z_eff_) and the comparable cross-sections of Pb and I (Figure 2f), which dominate the interaction probability regardless of cationic composition.

In this study we focus on the most probable interaction channels, whose cross-sections are of comparable magnitude. Rare, highly improbable events do occur but represent isolated cases with negligible statistical weight and therefore are not considered.

The photoelectric effect and ionization yields (Figure 3a–e) are nearly identical for all materials except the CsPbI3 perovskite, which shows a slightly higher ionization probability.

This trend can be attributed to the presence of Cs atoms, which increase the fraction of heavy nuclei in the composition. This interpretation is supported by the fact that Pb—present in all samples—has the highest photoelectric cross-section among the constituent elements (Figure 3f).

By contrast, the ionization metric used here aggregates secondary-electron production from all channels (photoelectric effect, Compton scattering, and at the higher energy range—pair production) and is therefore more sensitive to composition. Spatially, photoelectric absorption deposits energy more locally around the absorption site via short-range Auger and other cascades (especially at sub-MeV photon energies), whereas ionization by energetic secondaries produces longer, more delocalized electron tracks. The additional free electrons generated through ionization may initiate secondary processes such as redox-like reactions of Pb and halides, structural defect formation, or even partial decomposition of organic cations, all of which are central to long-term material degradation under irradiation.

Compton scattering (Figure 4a–e) appears nearly identical across all perovskites, which is consistent with their similar Z_eff_ values in the relevant energy range. In contrast, Rayleigh scattering is reduced for organic-cation-containing compositions (FA, MA), which have a higher fraction of light atoms and thus lower coherent scattering probability.

The cross-sections for Pb (Figure 4f) show the expected dominance of Compton scattering at intermediate energies and a smaller but noticeable contribution from Rayleigh scattering, explaining the trends observed in mixed compositions.

In GEANT4 simulations of γ-radiation interactions with single- and multi-cation perovskites, most processes exhibit minimal dependence on cationic composition, primarily because heavy atoms like Pb and I dominate interactions, present in comparable amounts across all materials. Deposited energy shows slight variations, being marginally higher in Cs-containing perovskites (e.g., CsPbI_3_) due to high atomic number of Cs enhancing absorption, while FA-rich compositions with more light atoms display lower values; NIEL and event rates remain negligible in differences, as γ-rays cause minimal atomic displacements and share similar Z_eff_. Ionization yields are nearly identical except for a slight increase in CsPbI_3_, attributed to Cs boosting the fraction of heavy nuclei and photoelectric cross-sections (highest for Pb), leading to more secondary electrons that could initiate degradation via redox reactions or defects. Photoelectric effects are uniform, but Rayleigh scattering is reduced in organic-cation (FA/MA) perovskites due to higher light-atom fractions lowering coherent scattering probability. Compton scattering, dominant at intermediate energies, shows negligible dependence on composition owing to comparable Z_eff_ and cross-sections. The unique absorption curves in perovskites feature a pronounced minimum from the interplay of photoelectric dominance transitioning to Compton and Rayleigh scattering, influenced by the high-Z/low-Z element mix.

To optimize the interaction efficiency in photon detection (to increase absorption and ionization for better signals), incorporate more heavy atoms, use thicker layers (≥1 mm), and favor inorganic compositions for higher Z_eff_. Conversely, for solar cells (to decrease absorption and ionization and minimize radiation damage, e.g., in space conditions), select FA/MA-rich perovskites to reduce absorption and ionization, avoid heavy cations, employ thin layers (<1 mm) with low-Z shields, and balance dopants to shift the absorption minimum toward cosmic ray energies, enhancing radiation hardness without compromising photovoltaic performance.

Overall, perovskites’ uniform γ-response highlights their versatility, with composition fine-tuning enabling application-specific adaptations, warranting further experiments on hybrids.

### 3.2. Energy Influence on Radiation Processes

To explore a broader energy domain (0.1–90 MeV), MAPbI_3_ was taken as a representative example to analyze the absorbed energy, energy per event, events per particle, and the relative contributions of the main interaction processes (Figure 5).

In the low-energy range (0.1–100 keV), the primary processes are photoelectric absorption and electron ionization (Figure 5b). Photons efficiently eject electrons from atomic shells, particularly in heavy elements such as Pb and I. This leads to a high number of events per particle (Figure 5a) and significant localized energy absorption, explaining the peak observed at low energies.

In the intermediate-energy range (100 keV–1 MeV), photoelectric absorption becomes less significant, while Compton and Rayleigh scattering dominate (Figure 5c). Photons are no longer fully absorbed but are partially scattered, transferring only a fraction of their energy to electrons. As a result, the number of events per particle decreases but does not fall to zero. The energy per event rises moderately, and the absorbed energy is distributed more deeply within the material. The light atoms (hydrogen, carbon, nitrogen) in the MA-cation contribute very little to these processes because of their low Z-number and small cross-sections at this energy range compared to lead and iodine. In contrast, the all-inorganic CsPbI_3_ exhibits a slightly higher number of events, consistent with its higher Z_eff_ due to cesium replacing the light organic cation.

In the high-energy range (>1.022 MeV), pair production becomes the dominant interaction channel (Figure 5d). The minimum in absorbed energy occurs in the transition region where neither the photoelectric effect nor Compton and Rayleigh scattering dominates. As the photon energy increases further, the probability of pair production rises. The resulting energetic electron and positron induce additional ionization cascades. The positron, before annihilation, may also scatter and lose energy via ionization. When annihilation finally occurs, the two 511 keV photons produced can undergo Compton scattering, multiple scattering, giving rise to secondary electrons and further energy deposition. This chain of interactions explains the increase in both the number of events (annihilation, multiple scattering) and the total absorbed energy at higher photon energies.

At the higher energies (>10 MeV), photo-nuclear reactions such as (γ, n), (γ, p), (γ, α), etc., may occur. Their cross-sections, however, are typically in the microbarn-to-millibarn range—orders of magnitude smaller than electromagnetic processes—so they play only a minor role in the overall ionization balance. In such cases, the photon excites the nucleus, which can then relax by re-emitting γ-radiation. This secondary γ-ray can transfer energy back into the electronic subsystem, indirectly contributing to ionization. However, the overall effect remains small compared to pair production, Compton- and multiple scattering-driven cascades.

Thus, the energy dependence of γ-ray interactions in MAPbI_3_ reflects a clear evolution of dominant processes: at low energies, the photoelectric effect governs; at intermediate energies, Compton and Rayleigh scattering dominate; and at high energies, pair production and the subsequent cascades of secondary ionization and annihilation events take over. In CsPbI_3_, the slightly higher interaction probability reflects the heavier Cs atom and a correspondingly larger Z_eff_. The interplay of these mechanisms, together with possible secondary cascades of radiation effects from energetic electrons, leads to multiple scattering events and deep, distributed energy deposition within the perovskite lattice. Appendix A provides detailed tables of the number of generated (Appendix A) and emerging (Appendix A) particles, as well as their energy distributions, which support these interpretations.

### 3.3. Layer Thickness Influence on Radiation Processes

To further investigate the radiation response of perovskite materials, the effect of thickness reduction was considered. Following the analysis of MAPbI_3_ in the broad photon energy range, we examined γ-ray interactions for perovskite layers with thicknesses of 1 mm, 100 μm, 10 μm, and 1 μm.

As expected, decreasing the thickness of the material reduces the effective interaction volume. Since γ-rays are weakly interacting and highly penetrating, especially at several MeV energies, the probability of absorption or scattering processes decreases accordingly. This (Figure 6b) results in a pronounced, non-linear drop in the number of events per particle when moving from 1 mm down to 1 μm. Interestingly, the 10 μm case shows slightly fewer events than the 1 μm case, while the slope of energy dependence is less steep. This may be due to statistical fluctuations of rare interaction events at intermediate thicknesses, where the absolute number of interactions is already very low.

The energy per event (Figure 6c) increases with decreasing thickness, reflecting the reduced number of total events: fewer interactions lead to each individual event carrying more weight in terms of deposited energy. For the 10 μm and 1 μm cases, the curves exhibit non-monotonic features and local extrema, which likely arise from the combination of statistical fluctuations and the discrete dominance of different interaction processes in narrow energy windows.

The total deposited energy (Figure 6a) systematically decreases with thickness, as a thinner target allows most photons to pass through without significant interaction. In thick perovskites, deposited energies reach several tens of keV in the 0.1–1 MeV range, while for 1 μm layers, the deposited energy falls to just a few tens of eV. This reflects the expected reduction in energy absorption with diminishing interaction volume.

Turning to individual processes, both the photoelectric effect (Figure 7a) and ionization (Figure 7b) show a decrease in event counts with increasing photon energy and decreasing thickness. Photoelectric absorption follows a predictable monotonic decline, consistent with its ~E^−3^ dependence. Ionization, however, shows an unexpected behavior: for 10 μm thickness, it is slightly lower than at 1 μm. This may be due to local variations in secondary electron generation and transport in extremely thin targets, where boundary effects and escape of secondaries play a larger role than in thicker volumes.

Compton (Figure 7c) and Rayleigh scattering (Figure 7d) scale more directly with thickness, showing an approximately order-of-magnitude reduction in event counts for each order-of-magnitude decrease in thickness.

For Compton scattering (Figure 7c), at the 1 mm curve, the energy spectrum exhibits a distinct maximum in the 0.15–3 MeV region before decreasing smoothly at higher energies. In contrast, for thinner samples (<1 mm), this pronounced peak vanishes, leaving only a smooth, monotonic decrease in event counts with increasing photon energy. This peak in the 1 mm case arises from the energy dependence of the Compton scattering cross-section itself. The cross-section (probability of interaction) of this process has a maximum for photon energies in the range of several hundred keV. In a thicker sample, this inherent property of the interaction probability is fully manifested, resulting in the observed maximum. For thinner perovskites, the total number of scattering events is drastically reduced, scaling linearly with thickness. This reduction uniformly lowers the count rate across all energies, effectively suppressing the visibility of the cross-section’s maximum and leaving only a smooth, decreasing trend dominated by the high-energy tail of the event distribution.

Rayleigh scattering (Figure 7d) falls off more rapidly than Compton (Figure 7c) across all thicknesses, which is consistent with its strong dependence on Z-number and the reduction in coherent scattering probability in thinner targets. This steeper decline is a direct consequence of the fundamental nature of the Rayleigh process. Unlike Compton scattering, which is an incoherent interaction with individual electrons, Rayleigh scattering is a coherent process where the photon wave interacts elastically with the entire atom. This coherence requires that the photon’s wavelength is comparable to the atomic spacing, making its cross-section peak sharply at lower photon energies and fall off precipitously with increasing energy (~E^−2^). Consequently, as sample thickness decreases, the already low probability of a coherent interaction is further diminished across the entire spectrum, leading to a rapid, uniform reduction in counts without the complex spectral features seen in the thicker-perovskite Compton data.

For the multiple scattering (Figure 7e), the results show a distinctive local minimum at 0.3–0.4 MeV for the 1 mm case, followed by a maximum near 1 MeV and then a gradual decline, are governed by the shifting dominance of photon interaction processes. The minimum corresponds to an energy region where the probability of a photon being absorbed (via the photoelectric effect) after one or more scatters is still significant, creating an extremum in the curve. As energy increases beyond this point into the Compton-dominated regime, the probability of absorption drops sharply. Photons can now undergo multiple scattering events without being fully absorbed, leading to an increased probability of being detected, which creates the observed maximum near 1 MeV. The final gradual decline reflects the decreasing of Compton scattering cross-section and increasing of pair-production and their annihilation process at higher energies.

For ultra-thin layers (10 μm and 1 μm), the interaction picture changes fundamentally. The concept of multiple scattering is no longer applicable; instead, the spectrum is shaped by a limited number of single-scattering events together with a dominant fraction of unscattered transmitted photons. The anomalously high counts at very low energies (around 0.1 MeV) arise mainly from this transmitted component of the primary beam, which passes through the target without interaction.

In this regime, the material response is largely determined by the transmitted photons, with only a small contribution from scattered radiation. The sharper decline observed for the 1 μm sample occurs because attenuation is negligible across most of the energy range. The transmitted spectrum therefore follows the source’s original energy distribution more closely and appears sharper, since there is no broadening from multiple interactions. In contrast, the 10 μm layer is thick enough to begin attenuating the lowest-energy transmitted photons, which makes the scattered component more noticeable and results in a less abrupt decline.

For thinner layers, this structure is less pronounced, and at 10 μm and 1 μm, the initial event counts at 0.1 MeV are anomalously higher, though the subsequent decline with energy is sharper for 1 μm. This can again be attributed to statistical effects and the stronger role of photon transmission versus scattering in ultra-thin layers.

Finally, pair production and the subsequent annihilation events (Figure 7f) appear only at photon energies above 1 MeV, consistent with the well-known threshold of 1.022 MeV corresponding to the combined rest mass of an electron–positron pair. At these energies, the photon is converted into an electron and a positron in the Coulomb field of a nucleus, followed by rapid annihilation of the positron into secondary photons.

The probability of observing such events decreases sharply with decreasing sample thickness. At 1 mm, pair-production signatures are still occasionally registered. At 100 μm, only very rare instances remain, and below this thickness no events are observed at all. This behavior reflects two fundamental aspects: the intrinsically low cross-section of pair production compared to other γ-interaction mechanisms, and the progressive reduction in the effective interaction volume as the material becomes thinner.

In ultra-thin layers, the physical path available for photons to interact is simply too short for such rare processes to occur with measurable frequency. As a result, pair production and annihilation make no contribution in the micrometer regime, and the overall γ-ray response is governed entirely by photoelectric effect, electron ionization and Compton and Rayleigh scattering.

These findings have direct implications for the design of perovskite-based devices. For radiation detectors, the choice of thickness provides a simple and effective way to tune sensitivity to different parts of the γ-ray spectrum. Ultra-thin layers in the micrometer range transmit most photons, making them unsuitable for efficient absorption but useful in transmission-mode detection, where minimal spectral distortion is desired. Intermediate thicknesses on the order of tens to hundreds of micrometers enable the resolution of characteristic scattering features, particularly in the 0.1–1 MeV range and may be attractive for compact detectors that balance efficiency with energy resolution. Millimeter-scale perovskite layers, by contrast, are required when the goal is to maximize absorption efficiency or to access high-energy processes such as pair production, making them suitable for γ-ray imaging or dosimetry at MeV energies.

In the context of perovskite solar cells, thickness control also plays a crucial role, although the relevant photon energies are orders of magnitude lower. For photovoltaic applications, the active layer is typically optimized in the nanometer to micrometer range to achieve efficient absorption of visible light while maintaining charge transport and minimizing recombination losses. Interestingly, this regime overlaps with the transmission-dominated γ-ray response discussed here, meaning that perovskite solar cells would be almost transparent to γ radiation and hence inherently radiation-hard under high-energy photon exposure. This duality underscores the versatility of MAPbI_3_: while thick crystals are required for photon detection, thin layers maintain photovoltaic performance even in harsh radiation environments, suggesting complementary applications of the same material in both detection technologies and space photovoltaics.

Because no systematic experimental reference exists for the γ-response of hybrid halide perovskites, we validated our GEANT4 implementation against materials with well-established radiation interaction data, namely aluminum, lead, and Mylar as a representative organic medium. For electrons, stopping powers and penetration ranges were calculated and compared to the NIST ESTAR and EXFOR databases, with agreement within the expected accuracy margins of the physics models.

It is important to note that GEANT4 employs a consistent set of electromagnetic interaction models for both electron and photon transport (covering ionization, scattering, and bremsstrahlung for electrons, and photoelectric effect, Compton scattering, Rayleigh scattering, and pair production for photons). Therefore, the good agreement observed in the electron case strongly suggests that the simulated γ-interaction trends in perovskites are likewise physically consistent.

In contrast to most experimental studies [37,38,39,40], which primarily probe macroscopic detector signals (charge collection, current response, radiation-induced degradation), our approach isolates the underlying microscopic interaction processes—photon absorption, ionization cascades, non-ionizing energy loss—as a first step. This process-level perspective complements signal-oriented measurements by providing mechanistic insight that cannot be directly extracted from current–voltage or spectroscopic readouts alone.

In the perovskite lattice, γ-irradiation interacts predominantly via discussed radiation effects, producing secondary electrons with energies far exceeding chemical bond strengths. These electrons can trigger the formation of halide vacancies, which act as traps induce non-radiative recombination and accelerate degradation under illumination. While the primary γ-rays do not directly displace ions, secondary cascades and, at higher energies, photonuclear events contribute to the defect population.

In addition, the relaxation of electronic excitations—including hot carriers, self-trapped excitons, and localized polarons—can couple to the soft ionic lattice and release sufficient energy to break Pb-I bonds. This non-thermal pathway further contributes to defect creation, complementing the direct electronic collision cascades.

From a materials perspective, iodine vacancies are particularly critical, as they serve as nucleation centers for further thermal and photochemical degradation. In proton-irradiated cells, such vacancies are distributed along the tracks [41]; under γ-irradiation, they are more uniformly generated by secondary electrons throughout the absorber.

A more detailed picture of vacancy formation and energy transfer into lattice vibrations requires coupling Monte-Carlo simulations with first principle calculations. Molecular dynamics and density functional theory calculations could provide quantitative insights into defect energetics, migration pathways, and the relative contribution of phonon heating versus carrier trapping.

## 4. Conclusions

The simulations reveal that the γ-radiation response of halide perovskites is governed primarily by the presence of heavy atoms (Pb and I), which dominate both photoelectric absorption and scattering processes. As a result, the overall absorbed energy and event rates are broadly similar across various compositions. Nevertheless, systematic differences emerge: Cs-containing perovskites show slightly higher absorbed energy and ionization yields, whereas FA- and MA-rich compositions, with a larger fraction of light atoms, exhibit reduced photoelectric absorption and Rayleigh scattering. These trends underline the role of the A-site cation in fine-tuning the balance of interaction processes, even if its effect is smaller than that of Pb and I atoms.

Thickness strongly modulates the radiation response—reducing the perovskite layer from the millimeter to the micrometer scale results in a pronounced, non-linear decrease in deposit energy and interaction probability. Thus, in ultrathin films, individual events carry more energy due to the reduced number of total interactions, statistical fluctuations become increasingly visible due to boundary effects, and the escape of secondary particles contribute disproportionately to the deposited energy distribution. Across the studied energy range (0.1–90 MeV), the expected sequence of dominant processes is observed: photoelectric effect at low energies, Compton and Rayleigh scattering at intermediate energies, and pair production above 1 MeV, with the latter suppressed in thin layers due to the limited interaction volume.

Together, these findings highlight two key aspects: (i) the γ-ray attenuation capacity of perovskites is determined by their heavy-atom content, while the choice of A-site cation fine-tunes the balance between competing processes; (ii) device performance in photon detection is critically dependent on perovskite thickness, with millimeter-scale volumes required for efficient γ-ray absorption, while micrometer-scale films remain relevant for applications relying on partial transmission, such as X-ray imaging and dosimetry. Thus, in this context, the cationic composition primarily affects the material’s optoelectronic properties and stability rather than its fundamental γ-interaction cross-sections, which are governed by Pb and I content.

## Figures and Tables

**Figure 1 nanomaterials-15-01474-f001:**
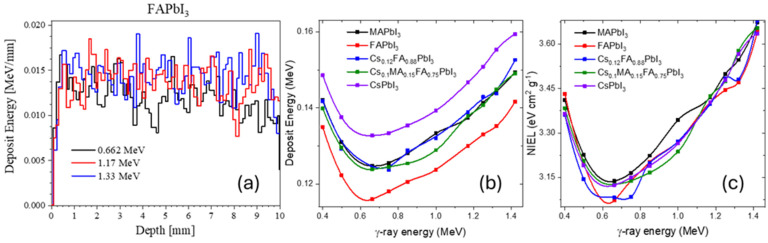
Simulated deposit energy as a function of perovskite thickness (**a**) for FAPbI_3_ at common calibration energies; deposit energy (**b**) and NIEL (**c**) for different halide perovskites.

**Figure 2 nanomaterials-15-01474-f002:**
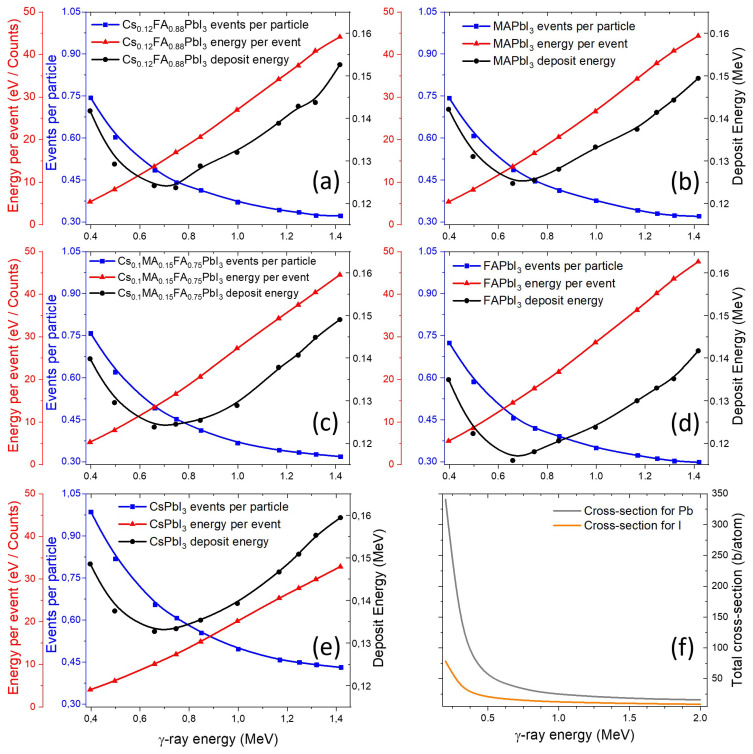
Calculated event rates, energy per event, and deposit energy for various perovskites under γ-ray irradiation (**a**–**e**), with Pb and I interaction total cross-sections (**f**).

**Figure 3 nanomaterials-15-01474-f003:**
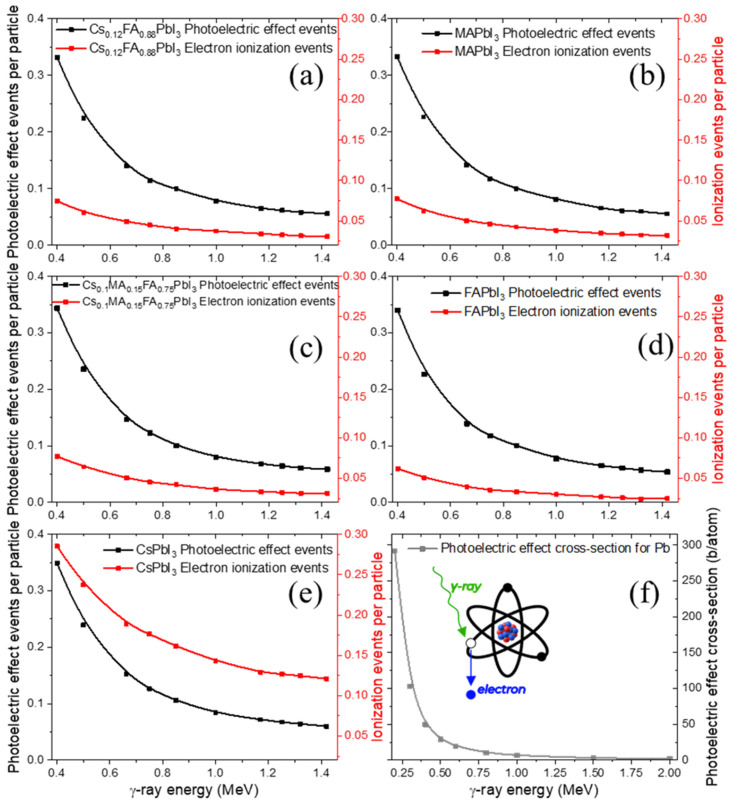
Calculated photoelectric effect and electron ionization event rates for various perovskites under γ-ray irradiation (**a**–**e**), with Pb photoelectric effect cross-section (**f**).

**Figure 4 nanomaterials-15-01474-f004:**
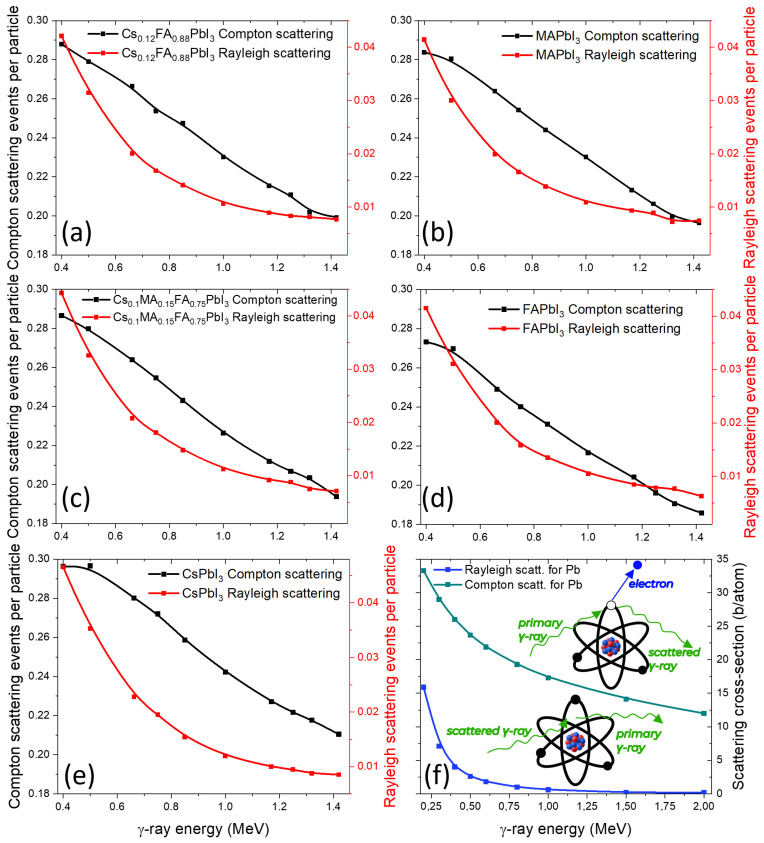
Calculated Compton and Rayleigh scattering event rates for various perovskites under γ-ray irradiation (**a**–**e**), with Pb Compton and Rayleigh scattering cross-sections (**f**).

**Figure 5 nanomaterials-15-01474-f005:**
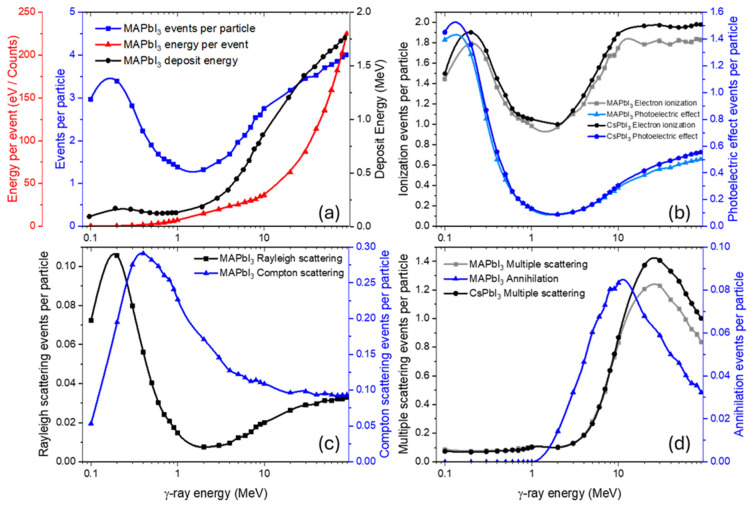
Energy dependence of γ-ray interaction characteristics (**a**) and dominant processes (**b**–**d**) in MAPbI_3_ (CsPbI_3_ curves are shown for comparison).

**Figure 6 nanomaterials-15-01474-f006:**
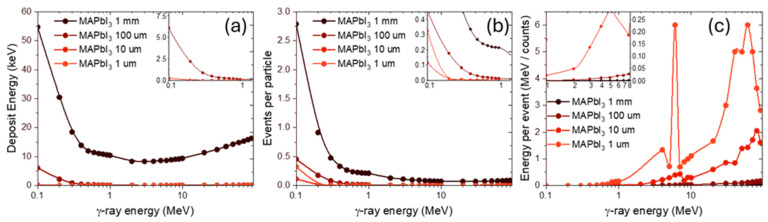
Energy dependence of γ-ray interaction characteristics in MAPbI_3_ at various thicknesses. (**a**) Deposit Energy; (**b**) Events per particle; (**c**) Energy per event.

**Figure 7 nanomaterials-15-01474-f007:**
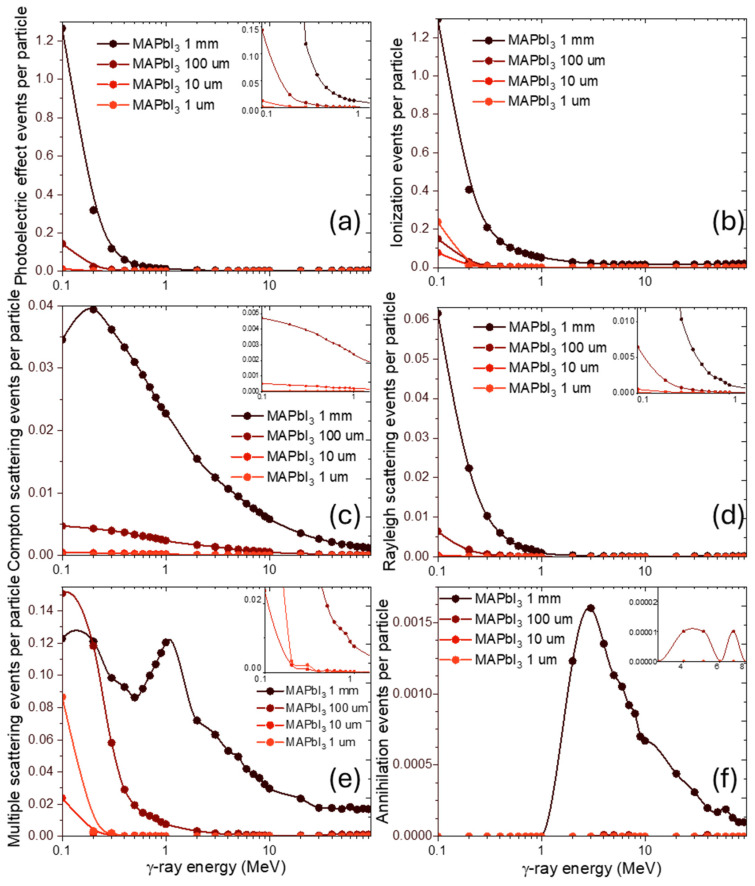
Energy dependence of dominant radiation-induced events in MAPbI_3_ at various thicknesses under γ-rays. (**a**) Photoelectric effect events per particle; (**b**) Ionization events per particle; (**c**) Compton scattering events per particle; (**d**) Rayleigh scattering events per particle; (**e**) Multiple scattering events per particle; (**f**) Annihilation events per particle.

**Table 1 nanomaterials-15-01474-t001:** Calculated γ-ray interaction characteristics of perovskites at reference photon energies.

Energy, MeV	Parameter	MAPbI_3_	FAPbI_3_	Cs_0.12_FA_0.88_PbI_3_	Cs_0.1_MA_0.15_FA_0.75_PbI_3_	CsPbI_3_
0.66	Deposit Energy, keV	122.52	119.75	121.71	123.83	132.86
	NIEL, eV∙cm^2^∙g^−1^	3.21	3.13	3.14	3.21	3.23
	Number of scattering events with energy loss	0.39	0.38	0.38	0.39	0.42
	Projective range, mm	9.18	9.22	9.21	9.19	9.12
1.17	Deposit Energy, keV	141.94	136.28	140.34	138.02	140.08
	NIEL, eV∙cm^2^∙g^−1^	3.61	3.45	3.51	3.47	3.29
	Number of scattering events with energy loss	0.27	0.27	0.27	0.26	0.27
	Projective range, mm	9.64	9.67	9.63	9.67	9.65
1.33	Deposit Energy, keV	139.84	141.37	151.70	139.83	151.12
	NIEL, eV∙cm^2^∙g^−1^	3.47	3.55	3.80	3.48	3.52
	Number of scattering events with energy loss	0.24	0.23	0.25	0.24	0.26
	Projective range, mm	9.74	9.75	9.71	9.75	9.70

## Data Availability

The data sharing is not applicable.

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
