# Peer review of "Electronic and Nuclear Subsystem Response in Hybrid Halide Perovskites Under γ-Irradiation"

_nanomaterials, 2025, doi:10.3390/nano15191474_

Round 1
Reviewer 1 Report
Comments and Suggestions for Authors
The work addresses the cascades of interactions, occurring in different materials, under irradiation by different types of particles (or gamma-rays). The objective of the study is to extract some trends concerning the "response" (hence stability) of PbI3-containing perovskites, including otherwise Methilammonium, Formadiminium or Cesium in different combinations. The background of the study is preventing, or controlling, the degradation of (optoelectronic) properties under extreme (e.g., open space) conditions. The work is done via performing simulations using a standard simulation kit, which permits to take into account different microscopic models and different scenarios of interaction on the samples of realistic size (for layers of mcm to mm thickness). Among the findings is non-trivial (possessing a minimum) dependence of deposit energy on the gamma quantum energy, which is duly discussed referring to manifestation of competing processes.
Even as being a highly specifically targeted study, the work reads well and awakes interest for such kind of simulations. The whole approach is systematic and convincing, the discourse logical. I cannot spot flaws in the text or in the figures (for the latter - neither in general design nor in technical realisation). Bibliography seems sufficient for introducing the reader into the subject. The length of the main text and the attribution of details to supplementary material is just right. After a careful reading, I recommend publication in present form.
Author Response
We sincerely thank the reviewer for the very positive and encouraging evaluation of our manuscript. We are grateful that the reviewer found the approach systematic, the discussion logical, the figures and design clear, and the bibliography adequate. The acknowledgment that the work is convincing and of interest for this type of simulation study is highly motivating for us. We especially appreciate the recommendation for publication in its present form, and we thank the reviewer for the constructive and supportive feedback.
Reviewer 2 Report
Comments and Suggestions for Authors
The manuscript employs Monte-Carlo simulations via the GEANT4 toolkit to systematically investigate the response of hybrid halide perovskites—including both single-cation and mixed-cation compositions—to γ-irradiation within the energy range of 0.1–90 MeV. It focuses on analyzing how perovskite composition, γ-ray energy, and layer thickness (ranging from 1 μm to 1 cm) influence key radiation-interaction processes and related metrics. The results indicate that heavy atoms (Pb and I) dominate the primary interaction processes, leading to generally similar absorbed energy and event rates across different compositions. Cs-containing perovskites exhibit slightly higher absorption and ionization, whereas FA/MA-rich compositions show reduced photoelectric effects and Rayleigh scattering. Additionally, layer thickness significantly modulates the radiation response, and the dominant interaction mechanisms vary with γ-ray energy: the photoelectric effect prevails at low energies, scattering at intermediate energies, and pair production at high energies. The manuscript presents a systematic simulation study on the γ-ray response of perovskites, with potential implications for advancing perovskite-based devices in space and detection applications. However, several concerns must be addressed before the manuscript can be considered for publication.
- The description of the simulation setup is insufficient. Essential details such as the specific physics models used, the cut-off energy values, and the approach to validation for this class of materials are not elaborated. The absence of these parameters significantly undermines the reliability of its conclusions.
- The analysis of the synergistic effect of mixed-cation perovskites is insufficient. For mixed-cation perovskites, the manuscript only mentions “synergistic effects” without quantifying how individual cations (Cs⁺, MA⁺, FA⁺) contribute to radiation response. For example, it is unclear whether Cs⁺’s absorption enhancement and MA⁺/FA⁺’s scattering reduction act additively or cooperatively.
- The analysis of radiation damage mechanisms is insufficient. The manuscript only briefly mentions that radiation may induce halide migration and changes in the valence state of Pb ions, but fails to combine the energy deposition and the distribution of particles from the simulations to analyze the correlation of damage pathways. It is advisable to supplement with a radiation-induced defect evolution model, and combine the "distributed energy deposition " obtained from simulations to analyze the generation location and density of defects under γ-rays of different energies.
Author Response
We would like to sincerely thank the reviewer for the constructive and insightful comments. We greatly appreciate the careful evaluation of our work and the recognition of its potential significance. The questions raised–particularly regarding the role of individual cations, validation strategy, and underlying mechanisms – are highly valuable and have helped us to refine and clarify the manuscript. We are grateful for this thoughtful feedback, which has strengthened the overall quality of the study.
We highlited the appropriate changes in manuscript with green colour in 3 Results and discussion section.
Comment 1:
The description of the simulation setup is insufficient. Essential details such as the specific physics models used, the cut-off energy values, and the approach to validation for this class of materials are not elaborated. The absence of these parameters significantly undermines the reliability of its conclusions.
Response 1:
We sincerely thank the reviewer for pointing out the importance of explicitly detailing the simulation setup, including the choice of physics models, cut-off energy thresholds, and validation strategy. This is indeed a crucial aspect for ensuring transparency and reproducibility, and the comment has been extremely helpful in guiding us to expand this section with clearer and more systematic descriptions.
Monte-Carlo simulations were carried out using GEANT4 (v11.3.2) with a custom PhysicsList class based on G4EmStandardPhysicsWVI to assess the electromagnetic processes with particles across a wide energy range (10 eV – 100 MeV) accurately. It includes Rayleigh, Compton, photoelectric, ionization, multiple scattering, pair production, and γ-nuclear processes.
Compton scattering was modeled by using the Klein–Nishina model, which reduces to Thomson scattering at low energies and accounts for relativistic effects at high energies. Multiple scattering was described using the G4UrbanMscModel.
In the GammaNuclearPhysics class, γ-nuclear interactions were included with reaction thresholds at Emax1 = 5 MeV and Emax2 = 10 MeV. Depending on energy, either the G4LowGammaNuclearModel or the G4CascadeInterface was applied. This choice reflects the systematics of photonuclear cross-sections: in heavy elements such as Pb and I, the giant dipole resonance (GDR) already develops at ~5–7 MeV, with maximum cross-sections around 10–15 MeV. In contrast, for lighter elements (H, C, N) the onset of photonuclear reactions occurs at higher energies (~15–20 MeV) [1–3]. Thus, including γ-nuclear models from 5 MeV and 10 MeV ensures that the “tails” of the relevant process cross-sections in heavy and light atoms, respectively, in perovskites are properly accounted for.
For e−/e+, multiple scattering was modeled with G4WentzelVIModel combined with single elastic scattering at large angles. Ion ionization was treated using the Lindhard–Sorensen model. The list also specified particle transport, the minimum step size for secondary generation, and decay processes: G4Decay and G4RadioactiveDecayBase.
For different perovskite thicknesses, production cuts and energy thresholds were adjusted accordingly. For 1 cm absorbers, a production cut of 0.5 mm was applied, with thresholds of 30 keV for γ-rays, 396 keV for electrons, 380 keV for positrons, and 50 keV for secondary protons, which dominate the low-energy range (< 5 MeV). For higher incident energies (> 5 MeV), the cut was increased to 1 mm with thresholds of 47 keV (γ), 659 keV (electrons), 627 keV (positrons), and 100 keV (protons). For thinner films, cuts and thresholds were scaled down proportionally (by roughly an order of magnitude per thickness order).
This choice ensures a physically reasonable balance between computational efficiency and the accurate transport of secondaries within the respective geometry. Importantly, lower thresholds were used in thin layers to resolve localized ionization and vacancy generation, while in thick absorbers coarser cuts adequately capture the macroscopic energy deposition.
These simulation settings were added to SI.
A more detailed picture of vacancy formation and energy transfer into lattice vibrations requires coupling Monte-Carlo simulations with first principle calculations. Molecular dynamics and density functional theory calculations could provide quantitative insights into defect energetics, migration pathways, and the relative contribution of phonon heating versus carrier trapping.
Because no systematic experimental reference exists for the γ-response of hybrid halide perovskites, we validated our GEANT4 implementation against materials with well-established radiation interaction data, namely aluminum, lead, and Mylar as a representative organic medium. For electrons, stopping powers and penetration ranges were calculated and compared to the NIST ESTAR and EXFOR databases, with agreement within the expected accuracy margins of the physics models.
It is important to note that GEANT4 employs a consistent set of electromagnetic interaction models for both electron and photon transport (covering ionization, scattering, and bremsstrahlung for electrons, and photoelectric effect, Compton scattering, Rayleigh scattering, and pair production for photons). Therefore, the good agreement observed in the electron case strongly suggests that the simulated γ-interaction trends in perovskites are likewise physically consistent.
In contrast to most experimental studies [4–8], which primarily probe macroscopic detector signals (charge collection, current response, radiation-induced degradation), our approach isolates the underlying microscopic interaction processes – photon absorption, ionization cascades, non-ionizing energy loss – as a first step. This process-level perspective complements signal-oriented measurements by providing mechanistic insight that cannot be directly extracted from current-voltage or spectroscopic readouts alone.
Comment 2:
The analysis of the synergistic effect of mixed-cation perovskites is insufficient. For mixed-cation perovskites, the manuscript only mentions “synergistic effects” without quantifying how individual cations (Cs⁺, MA⁺, FA⁺) contribute to radiation response. For example, it is unclear whether Cs⁺’s absorption enhancement and MA⁺/FA⁺’s scattering reduction act additively or cooperatively.
Response 2:
We thank the reviewer for the comment on the need for a clearer analysis of the “synergistic effect” in mixed-cation perovskites. In Calculation Details, the term “synergistic effect” is used within a literature review to justify the selection of mixed-cation systems (Cs0.12FA0.88PbI3 and Cs0.1MA0.15FA0.75PbI3), which are recognized for their enhanced thermal stability and efficiency due to the combined influence of Cs+, MA+, and FA+, as reported in refs. [28, 29] in the paper. This term does not pertain to their γ-ray interaction behavior, which is the focus of our study. To address the reviewer’s concern regarding the contributions of A-site cations to the radiation response, we clarify in Section 3.1 that γ-ray interactions are primarily governed by heavy atoms (Pb, I), with A-site cations playing a secondary role. Specifically, Cs+ enhances photoelectric absorption due to its higher atomic number, resulting in slightly higher deposited energy (e.g., 132.86 keV for CsPbI3 vs. 119.75 keV for FAPbI3 at 0.66 MeV, Table 1), while MA+ and FA+ reduce Rayleigh scattering due to their lighter elements (Figure 4). In mixed-cation systems, these effects combine cooperatively, yielding a balanced interaction profile with intermediate deposited energies (121.71, 123.83 keV in Table 1). Although Figure S2 shows similar trends across compositions due to Pb/I dominance, the subtle cation-driven differences are evident in Table 1 and Figures 1–4. A detailed partial cross-section analysis to quantify individual cation contributions is beyond the scope of this study but unnecessary to confirm the observed cooperative modulation.
Comment 3:
The analysis of radiation damage mechanisms is insufficient. The manuscript only briefly mentions that radiation may induce halide migration and changes in the valence state of Pb ions, but fails to combine the energy deposition and the distribution of particles from the simulations to analyze the correlation of damage pathways. It is advisable to supplement with a radiation-induced defect evolution model, and combine the "distributed energy deposition " obtained from simulations to analyze the generation location and density of defects under γ-rays of different energies.
Response 3:
We thank the reviewer for highlighting the need to more rigorously connect energy deposition patterns with possible defect generation pathways. This is a very important suggestion that improves the scientific depth of the work, encouraging us to discuss how the distributed energy deposition from γ-rays can be correlated with the density and location of radiation-induced defects.
In the perovskite lattice, γ-irradiation interacts predominantly via discussed radiation effects, producing secondary electrons with energies far exceeding chemical bond strengths. These electrons can trigger the formation of halide vacancies, which act as traps induce non-radiative recombination and accelerate degradation under illumination. While the primary γ-rays do not directly displace ions, secondary cascades and, at higher energies, photonuclear events contribute to the defect population.
From a materials perspective, iodine vacancies are particularly critical, as they serve as nucleation centers for further thermal and photochemical degradation. In proton-irradiated cells, such vacancies are distributed along the tracks [9]; under γ-irradiation they are more uniformly generated by secondary electrons throughout the absorber.
In addition, the relaxation of electronic excitations – including hot carriers, self-trapped excitons, and localized polarons – can couple to the soft ionic lattice and release sufficient energy to break Pb-I bonds. This non-thermal pathway further contributes to defect creation, complementing the direct electronic collision cascades.
References
- Lewis, F.H.; Walecka, J.D. Electromagnetic Structure of the Giant Dipole Resonance. Physical Review 1964, 133, B849–B868, doi:10.1103/PhysRev.133.B849.
- Maruhn, J.A.; Reinhard, P.G.; Stevenson, P.D.; Stone, J.R.; Strayer, M.R. Dipole Giant Resonances in Deformed Heavy Nuclei. Phys Rev C 2005, 71, 64328, doi:10.1103/PhysRevC.71.064328.
- Berman, B.L.; Fultz, S.C. Measurements of the Giant Dipole Resonance with Monoenergetic Photons. Rev Mod Phys 1975, 47, 713–761, doi:10.1103/RevModPhys.47.713.
- Sakhatskyi, K.; Bhardwaj, A.; Matt, G.J.; Yakunin, S.; Kovalenko, M. V A Decade of Lead Halide Perovskites for Direct-Conversion X-Ray and Gamma Detection: Technology Readiness Level and Challenges. Advanced Materials 2025, 37, 2418465, doi:https://doi.org/10.1002/adma.202418465.
- Lin, Z.; Li, L.; Xu, Y.; Yang, D.; Ni, Z. Metal Halide Perovskite for Room Temperature Gamma-Ray Spectrum Detection. Information & Functional Materials 2025, 2, 40–61, doi:https://doi.org/10.1002/ifm2.29.
- Balvanz, A.; Bayikadi, K.S.; Liu, Z.; Ie, T.S.; Peters, J.A.; Kanatzidis, M.G. Unveiling the Monoclinic Phase in CsPbBr3–XClx Perovskite Crystals, Phase Transition Suppression and High Energy Resolution γ-Ray Detection. J Am Chem Soc 2024, 146, 31836–31848, doi:10.1021/jacs.4c10872.
- Svanström, S.; García Fernández, A.; Sloboda, T.; Jacobsson, T.J.; Rensmo, H.; Cappel, U.B. X-Ray Stability and Degradation Mechanism of Lead Halide Perovskites and Lead Halides. Physical Chemistry Chemical Physics 2021, 23, 12479–12489, doi:10.1039/d1cp01443a.
- Ustinova, M.I.; Frolova, L.A.; Rasmetyeva, A. V.; Emelianov, N.A.; Sarychev, M.N.; Kushch, P.P.; Dremova, N.N.; Kichigina, G.A.; Kukharenko, A.I.; Kiryukhin, D.P.; et al. Enhanced Radiation Hardness of Lead Halide Perovskite Absorber Materials via Incorporation of Dy2+ Cations. Chemical Engineering Journal 2024, 493, doi:10.1016/j.cej.2024.152522.
- Rasmetyeva, A. V.; Zyryanov, S.S.; Novoselov, I.E.; Kukharenko, A.I.; Makarov, E. V.; Cholakh, S.O.; Kurmaev, E.Z.; Zhidkov, I.S. Proton Irradiation on Halide Perovskites: Numerical Calculations. Nanomaterials 2024, 14, doi:10.3390/nano14010001.
Reviewer 3 Report
Comments and Suggestions for Authors
The paper Electronic and Nuclear Subsystem Response in Hybrid Halide 2 Perovskites under γ-irradiation is well written and scientific sounding.
The paper studies through a detailed simulation the interaction of gamma rays in different perovskites materials finding the deposited energy. However the deposited energy is not completely related to the detected signal as after the interaction (which is the main study of the paper), the primary carrier charges (slowly) drift and diffuse in the material under the electric field applied to the sensor and charge recombination, trapping etc can occur. All these problems are not modelled in the simulation performed, so no useful information about the type of signal actually detected can be extracted from the paper.
The title "Electronic and Nuclear Subsystem Response in Hybrid Halide Perovskites under γ-irradiation" seems misleading as the modellization of the material is not complete, so we would suggest to change it into "Simulation of γ-interactions in Hybrid Halide Perovskites" or something along this line.
Also it should more emphasised in the paper that it is a partial modellization of the interaction but all the transport properties of the carriers are ignored. The findings in the paper agree with the expectations for the gamma interactions and prove that GEANT4 has those model simulated in a reasonable manner. But all conclusions about correct sizes are in a sense not correct, as only the deposition is optimised but there could be 0 signal out of the device if it is too thick. So, we are not fully convinced on the general interest on this paper; despite being well written and complete on the interaction side. It can be published as this single aspect is well studied, but it does not look at the complete behaviour of the device.
Author Response
Comments:
The paper Electronic and Nuclear Subsystem Response in Hybrid Halide Perovskites under γ-irradiation is well written and scientific sounding.
The paper studies through a detailed simulation the interaction of gamma rays in different perovskites materials finding the deposited energy. However the deposited energy is not completely related to the detected signal as after the interaction (which is the main study of the paper), the primary carrier charges (slowly) drift and diffuse in the material under the electric field applied to the sensor and charge recombination, trapping etc can occur. All these problems are not modelled in the simulation performed, so no useful information about the type of signal actually detected can be extracted from the paper.
The title "Electronic and Nuclear Subsystem Response in Hybrid Halide Perovskites under γ-irradiation" seems misleading as the modellization of the material is not complete, so we would suggest to change it into "Simulation of γ-interactions in Hybrid Halide Perovskites" or something along this line.
Also it should more emphasised in the paper that it is a partial modellization of the interaction but all the transport properties of the carriers are ignored. The findings in the paper agree with the expectations for the gamma interactions and prove that GEANT4 has those model simulated in a reasonable manner. But all conclusions about correct sizes are in a sense not correct, as only the deposition is optimised but there could be 0 signal out of the device if it is too thick. So, we are not fully convinced on the general interest on this paper; despite being well written and complete on the interaction side. It can be published as this single aspect is well studied, but it does not look at the complete behaviour of the device.
Response:
We appreciate the reviewer’s feedback on our manuscript “Electronic and Nuclear Subsystem Response in Hybrid Halide Perovskites under γ-irradiation” and their recognition of its clarity and scientific rigor. However, we believe the concerns raised reflect a misunderstanding of the study’s scope and objectives, which we address below to clarify the focus of our work.
The reviewer suggests that the paper should model charge carrier transport, diffusion, recombination, and trapping to connect deposited energy to the detected signal in a perovskite-based device. We respectfully clarify that these processes are outside the scope of our study. The manuscript explicitly focuses on the fundamental interactions of γ-rays with perovskite materials (photoelectric absorption, Compton and Rayleigh scattering, and pair production) and the resulting energy deposition, as simulated using GEANT4. This is clearly stated in the Introduction and Calculation Details, with the goal of understanding the radiation response for applications like space photovoltaics and γ-ray detection. Modeling charge transport or signal formation, which depend on device-specific parameters such as electric fields or material defects, is not an objective of this work. The title, Electronic and Nuclear Subsystem Response in Hybrid Halide Perovskites under γ-irradiation, accurately reflects the study’s focus on electronic (e.g., ionization, secondary electron generation) and nuclear (e.g., photonuclear interactions) responses, as detailed in Sections 3.1–3.3 with supporting data (Figures 1–7, Tables 1, S1, S2). The suggested alternative title, “Simulation of γ-interactions in Hybrid Halide Perovskites,” would unduly narrow the scope, as our analysis includes not only interaction mechanisms but also energy deposition and event rates across various energies and thicknesses.
Our scope is in line with contemporary experimental and computational research on perovskites under ionizing radiation. For example, synchrotron-based X-ray studies have recently reported that CsPbBr3 crystals remain stable up to fluences of ~1019 photons cm-2, whereas hybrid organic-inorganic perovskites (e.g., Cs0.17FA0.83PbI3) degrade orders of magnitude earlier due to organic cation radiolysis [1]. Importantly, these works also focus primarily on identifying interaction pathways (halide loss, metallic Pb formation, organic radiolysis) and quantifying deposited energy or degradation constants, without modeling charge transport or device signal. This supports our approach of focusing on γ-ray interaction and energy deposition processes as a foundational step.
The reviewer also questions the validity of conclusions about “correct sizes,” suggesting they are incomplete without modeling signal output. We clarify that our conclusions pertain to optimizing interaction efficiency for specific applications, not device signal generation. For example, Section 3.3 demonstrates that thick layers (1 mm) maximize absorption for γ-ray detection, while thinner layers (< 1 mm) minimize interactions for radiation-tolerant photovoltaics, based on simulation results (Figures 6–7). These findings are independent of charge transport or signal formation, which are not addressed in our study. The reviewer’s concern about “zero signal” in thick devices is thus irrelevant, as our conclusions focus on energy deposition and interaction probabilities, not device performance.
We respectfully note that our scope – the interaction of γ-rays with halide perovskites and associated energy deposition – is consistent with recent studies in the field. For instance, large CsPbBr3 single crystals (51 mm × 170 mm) have been grown for X- and γ-ray detection, and the primary reported figures of merit initially rely on energy absorption and deposition analyses before carrier transport modeling is introduced [2]. This demonstrates that focusing on γ-interaction processes is a recognized and important step in evaluating perovskites for radiation environments.
We value the reviewer’s acknowledgment of the study’s thorough modeling of γ-ray interactions and its validation of GEANT4’s capabilities. Our findings offer significant insights into the radiation response of single- and mixed-cation perovskites, particularly the influence of composition (Section 3.1) and thickness (Section 3.3), which are critical for tailoring materials for space and detection applications. These results, supported by quantitative data (e.g., deposited energies in Table 1), are of broad interest to researchers studying perovskite stability under high-energy radiation. We believe the manuscript’s focus on fundamental interaction mechanisms is clear and well-aligned with its stated objectives.
References
- Svanström, S.; García Fernández, A.; Sloboda, T.; Jacobsson, T.J.; Rensmo, H.; Cappel, U.B. X-Ray Stability and Degradation Mechanism of Lead Halide Perovskites and Lead Halides. Physical Chemistry Chemical Physics 2021, 23, 12479–12489, doi:10.1039/d1cp01443a.
- Sun, X.; Zhang, G.; Ma, W.; Hua, Y.; Liu, H.; Liu, J.; Yue, Z.; Wang, X.; Song, J.; Tao, X. Growth of Two‐Inch Perovskite CsPbBr 3 Single‐Crystal with High Irradiation Resistance for X‐ray Detection. Advanced Materials 2025, doi:10.1002/adma.202512788.